# Kinetic Monte Carlo Simulation of Clustering in an Al-Mg-Si-Cu Alloy

**DOI:** 10.3390/ma14164523

**Published:** 2021-08-12

**Authors:** Qilu Ye, Jianxin Wu, Jiqing Zhao, Gang Yang, Bin Yang

**Affiliations:** 1Collaborative Innovation Center of Steel Technology, University of Science and Technology Beijing, Beijing 100083, China; b20180505@xs.ustb.edu.cn (Q.Y.); wujx9@126.com (J.W.); 2Institute for Special Steels, Central Iron and Steel Research Institute, Beijing 100081, China; zhaojiqing@nercast.com (J.Z.); yanggang@nercast.com (G.Y.)

**Keywords:** aluminum alloy, Kinetic Monte Carlo, reversion, aging, activation energy

## Abstract

The mechanism of the clustering in Al-Mg-Si-Cu alloys has been a long-standing controversial issue. Here, for the first time, the mechanism of the clustering in the alloy was investigated by a Kinetic Monte Carlo (KMC) approach. In addition, reversion aging (RA) was carried out to evaluate the simulation results. The results showed that many small-size clusters formed rapidly in the early stages of aging. With the prolongation of aging time, the clusters merged and grew. The small clusters formed at the beginning of aging in Al-Mg-Si-Cu alloy were caused by initial vacancies (quenching vacancies). The merging and decomposition of the clusters were mainly caused by the capturing of vacancies, and the clusters had a probability to decompose before reaching a stable size. After repeated merging and decomposition, the clusters reach stability. During RA, the complex interaction between the cluster merging and decomposition leaded to the partial irregular change of the hardness reduction and activation energy.

## 1. Introduction

With the development of light-weight automobiles in recent years, aluminum alloys (especially 6000-series aluminum alloys) have become widely used to produce automobile body sheets due to their good formability, corrosion resistance and baking hardening [1]. The precipitation sequence of an Al-Mg-Si-Cu alloy is often as follows [2,3]:SSSS→atomic clusters→GP zones→β″, L/S/C, QP, QC→β′, Q′→Q, Si 
where SSSS is a supersaturated solid solution, GP zones are Guinier Preston zones, β″ phase is Mg_5_Si_6_ with monoclinic structure, L/S/C, QP and QC phases are Q′ precursors, β′ is Mg_9_Si_5_ with hexagonal structure, Q′ is probably Al_3_Cu_2_Mg_9_Si_7_ with hexagonal structure, Q is probably Al_3_Cu_2_Mg_9_Si_7_ with hexagonal structure.

6000-series aluminum alloys are age-hardening alloys. The early aging in Al-Mg-Si-Cu alloys, which directly affects the mechanical properties of these alloys by forming a main strengthening β″ phase, is associated with the aggregation of solute atoms (clusters) without structural changes [4]. Research on the aging behavior of clusters can provide ways for improving the properties of the alloy. However, due to the rapid aging, low contrast of solute atoms and metastable clusters, the early aging behavior of the alloy has been a longstanding controversial issue [5,6]. At present, atom probe tomography (APT) is the sole technique available for gaining direct insight into the early behavior of metallic systems [7]. However, the samples prepared for APT are a few hundred nanometers in size. Moreover, the detection efficiency of atoms mainly depends on the advanced level of the APT instruments. In the past, the detection efficiency was usually 37% to 57%, although recently, it has been increased to 80% for advanced instruments such as the CAMECA LEAP 5000 [8]. This means that only a fraction of atoms can be involved and there was a loss of lattice information in the reconstruction for most studies [9,10].

Computer simulations are helpful to understand the early precipitation of binary and more complex alloy systems 11. Many studies have shown that vacancies have a great influence on clustering in Al alloys [5,6,8,9,10]. During the early aging period, the solute atoms occupy the vacancies and form clusters through the mechanism of vacancy random walk [11]. The recently developed Kinetic Monte Carlo (KMC) method is mainly used to quantitatively study the evolution of group configuration and structure function with time during early aging by simulating the random walk of vacancies [12]. KMC is a statistical probability method which can study the random behavior of molecular chains deeply into the phase space, and solve the problem of complex system by repeated random sampling. It combines statistical physics to establish the relationship between the state of elementary particles and the macroscopic properties of materials. This method not only reveals the basic principles behind precipitation, but also the relationship between transformation and time. It can accurately simulate the microstructural evolution and diffusion behavior of an alloy, including nucleation, growth, phase separation dynamics, vacancy capture, diffusion mechanism, stability of small clusters and the role of microalloy elements in phase decomposition [13].

Reverse ageing (RA) treatment is another possible way to study the early aging behavior, in which a high enough temperature is used to dissolve the previously formed clusters instead of directly observing their formation after quenching. In this way, some of the solute atoms and the captured vacancies confined in the cluster can be returned to the matrix by RA. At the same time, one can derive properties of the clusters by investigating RA [14].

At present, KMC simulations have been carried out for Al-Mg, Al-Cu-Mg, Al-Mg-Zn-Cu, Al-Li and Al-Li-Cu [11,15,16,17]. However, the mechanism of clustering in Al-Mg-Si-Cu alloy is still controversial [5,6]. For example, the roles of quenching vacancies and capturing vacancies are unclear. Although both atomic simulation and atomic scale characterization have been developed recently, most of the studies on the microstructure simulation by KMC method have focused on the size distribution characteristics of the microstructure, while there is a lack of direct comparison between simulation and experimental results. Therefore, in this paper, the clustering during early aging of an Al-Mg-Si-Cu alloy was investigated by KMC and RA in order to understand the mechanism of clustering and provide theoretical guidance and technical support for making heat treatment process of the aluminum alloy.

## 2. Materials and Methods

The alloy simulated in this study was Al-0.45Mg-1.06Si-0.05Cu (weight %). The KMC in this work was mainly used to simulate the random walk of vacancies based on the exchange probability between vacancies and surrounding atoms. The initial KMC configuration is usually a single-phase structure obtained by quenching, so the initial model with 25 × 25 × 25 super-cell was established according to the composition and the vacancy concentration *C_V_* of the alloy quenched from 803 K to room temperature, and *C_V_* was calculated by Equation (1) [18]:(1)$$C_{V}=\exp(-\frac{G_{V}^{f}}{k_{B}T})$$
where *k_B_* is the Boltzmann constant and *T* is the solution temperature. *G_V_^f^* is the vacancy formation energy, which could be calculated by Equation (2) [19]:(2)$$G_{V}^{f}=H_{V}^{f}-TS_{V}^{f}$$

For 6000-series aluminum alloys, the vacancy formation enthalpy $H_{V}^{f}$ = 0.67 eV and formation entropy $S_{V}^{f}$ = 0.7 *k_B_* [19].

The diffusion behavior of atoms is realized by the exchange of their nearest vacancies. The Kinetic Monte Carlo simulation flow is shown in Figure 1a. The clusters containing more than two or three atoms can be automatically selected by the program, as shown in Figure 1b. Figure 1c shows the exchange behavior between vacancy and atoms.

At the beginning of the simulation, a vacancy was selected randomly, as shown in Figure 1a. Then, the energy differences Δ*E* between the vacancy and the 12 nearest neighboring atoms (the coordination number *z* is 12 for the aluminum alloy with FCC structure) before and after the exchange were calculated by the interatomic interaction potential (L-J potential), the parameters were given in Table 1 and Table 2 [11,20,21,22,23]. The interatomic and interatom-vacancy interaction potential parameters in Table 2 are derived from the parameters in Table 1 and the details of calculation were described in [11]. According to Equations (3)–(5) 11, Δ*E_a_* was calculated from Δ*E*, and atomic migration rate *w_i_* was calculated from atomic migration frequencies *v**_i_* and Δ*E_a_*. From Equation (7), one can see that atomic migration probability *α_i_* was calculated from *w_i_*. R in Equation (3) is the perfect gases universal constant, 8.314 J/(mol·K). If *a_i_* satisfies the condition in Figure 1a, the vacancy and the *k*-th atom will exchange to form a new distribution. In order to get a definite result, it usually takes more than 1 × 10^8^ Monte Carlo steps (MCS). Calculations were performed using a server with Core I7 processors at 1 × 10^5^ MCS per minute. Because each MCS of the computation depended on the results of the previous MCS, the program was serial, so only one CPU core was used for the computation. It took ~7 days to take 1 × 10^9^ MCS in this work. Atomic migration frequency *v_i_* was calculated from atomic diffusion constant *D_0_* and lattice constant *a*_0_ of pure metals, as shown in Equation (6). However, the temperature at which *D*_0_ was measured in [20,21,22,23] was above 520 K, so this value of *D_0_* was not suitable for natural aging (NA) (298 K). There would be an order of magnitude difference between *D*_0_ at 298 K and *D*_0_ at 520 K. From Equations (3), (6) and (7), one can see that this difference does not affect the calculation of *α**_i_* because the differences in order of magnitude cancel out each other (*v**_i_*/*v**_j_* is involved in the calculation) during the calculation. However, the atomic residence time *τ* and real physical time *t* are proportional to the reciprocal of the atomic migration frequency *w_i_*, as shown in Equations (8) and (9), in which MCS means Monte Carlo Step, *C_v_^0^* is the vacancy concentration used in the simulation, and *C_v_^q^* is the equilibrium vacancy concentration calculated from Equation (1). This means that the atomic diffusion constant *D*_0_ measured above 520 K will result in the unreasonable simulation time at a lower simulation temperature. Thus, at different simulated temperatures, *D*_0_ needs to be corrected in combination with the actual physical time of the experiment, at least in the order of magnitude:(3)$$w_{i}=\nu_{i}\exp(-\Delta E_{a}/RT)$$
(4)$$\Delta E_{a}=\Delta E/2+e_{0}$$
(5)$$\begin{array}{ll} e_{0} & =Q_{i}-\Delta H_{v}^{f}\;(\mathrm{in}\;\mathrm{Al}-i\;\mathrm{alloys})\\ & =Q_{i}-\left[\Delta H_{v}^{f}\;(\mathrm{in}\;\mathrm{pure}\;\mathrm{Al})-E_{i-v}^{b}\right] \end{array}$$
(6)$$\nu_{i}=D_{0}/a_{0}^{2}$$
(7)$$\alpha_{i}=w_{i}/\sum_{j=1}^{z}w_{j}$$
(8)$$\tau ={\left\{\sum_{j=1}^{z}w_{j}\right\}}^{-1}$$
(9)$$t=\left(\sum_{}^{MCS}\tau \right)/\left(C_{v}^{0}/C_{v}^{eq}\right)$$

The material used in the RA experiments was a cold-rolled sheet with 1 mm thickness. The samples were cut as discs of Φ4 mm × 1 mm and plates of 10 × 10 × 1 mm for differential scanning calorimetry (DSC) and hardness measurements, respectively, and then grounded and polished for DSC and hardness measurements, respectively. All samples were solutionised at 803 K for 1 h, quenched in water, some of them were directly naturally aged for 2 weeks at 293 K (T4 temper), and the others were treated in an oven at 343 K for 4 h and then naturally aged for 2 weeks at 293 K (T4P temper). RA was performed in PMX-200 silicone fluid held at 523 K in an oil-bath pot. Thermal analysis was performed using an EXSTAR6220 DSC (Seiko, Tokyo, Japan). The DSC chamber was pre-cooled to 273 K prior to placing a sample inside, and then DSC runs at a constant heating rate of 10 K·min^−1^ were performed from 273 K to 673 K after 2 min of equilibration. Vickers hardness was measured by applying a load force of 200 g. Ten measurements were taken for each sample.

## 3. Results and Discussion

### 3.1. Early Aging Behavior Simulation

In order to obtain a reasonable real physical time, it is necessary to adjust the diffusion constant *D*_0_ for natural aging (NA) (298 K), while the diffusion constant *D*_0_ for artifact aging (AA) (448 K) does not need to be adjusted because 448 K is close to the reported temperature ~500 K [20,21,22,23] for measuring the diffusion constant. The diffusion constant *D*_0_ was adjusted through the difference between the simulation time and the actual experiment time when cluster volume changes during natural aging. The first change is the rapid increase of cluster volume fraction, and the second change is the stabilization of cluster volume fraction. For 6000-series aluminum alloys, the hardness of a quenched sample usually increases rapidly within the first 2 h after quenching, and then enters a gentle transition stage until it stabilizes within 2 weeks (336 h) 5. There is a positive correlation between the hardness and the cluster volume fraction during NA, the hardness increases with the increase of the volume fraction of clusters [24,25,26]. Figure 2a shows the KMC simulation result of the cluster volume fraction (expressed by the number of solute atoms) during NA. One can see the cluster volume increased rapidly within the first 20 s. From the values simulated for 20 s and experimentally tested for 2 h, the corresponding coefficient (2 h/20 s = 360) of the time *t* for NA can be obtained, then the corresponding preexponential coefficient (360^−1^ ≈ 2.78 × 10^−3^) of diffusion can be obtained according to Equations (6–9). With the preexponential coefficient, the cluster volume fraction just tended to be stable within 300 h (about 2 weeks) during NA, and it is a reasonable real physical time, as shown in Figure 2b. During AA, the cluster volume fraction of the alloy tended to be stable for several minutes, as shown in Figure 2c. In the actual experiment, the best AA time of the alloy was about 20~30 min, because the β″ phase, transformed from the clusters, fully formed at this time. The β″ phase took only a few tens of minutes to form fully. Thus, the clusters might take only a few minutes to form fully. This means that the AA simulation without changing the preexponential coefficient of diffusion can get a reasonable real physical time *t* for AA. In fact, the preexponential coefficient of diffusion may be different in different simulations because the simulation time can be only consistent with the actual time in the order of magnitude. Although the preexponential coefficient of diffusion is different, the order of magnitude is similar, and the purpose is to make a better comparison between simulations and experiments.

Figure 3 and Figure 4 show the KMC simulation results for NA and AA of Al-0.45Mg-1.06Si-0.05Cu aluminum alloy, respectively. Except the maximum size of the clusters in AA samples, these simulation results are in good agreement with the 3D atom probe tomography (3DAPT) experimental results reported by Cao et al. 27. One can see that the maximum size of the clusters in NA samples was about 28 atoms (21 atoms measured by Cao et al.), and the average size was about six atoms (5.5 atoms measured by Cao et al.), as shown in Figure 3c. For AA, the maximum size of the cluster was about 80 atoms (212 atoms measured by Cao et al.), and the average size was about 12 atoms (11.5 atoms measured by Cao et al.), as shown in Figure 4c. In the actual experiment, the clusters with more than ~75 atoms can be called β″ phase [27,28,29]. There was a structural change during the transformation of clusters to β″ phase. It may be a complicated process, which required more complex steps to simulate the formation of β″ phase. The current molecular dynamics simulations with complex steps can only simulate the actual experimental process for a very short time (a few seconds at most). Therefore, it is not suitable for long time scale simulation, such as NA and AA in aluminum alloy. The KMC simulation in this work, which generally considered the nearest neighbor atomic interaction and cannot consider structural change, was suitable for investigating the behavior of the cluster rather than that of the β″ phase. Thus, the maximum size of the clusters in AA samples (80 atoms) was reasonable. According to the APT results reported by Cao et al. [27], the volume fraction of small aggregates with 4–9 Mg + Si + Cu atoms increased with increasing natural ageing time, while the volume fraction of aggregates with 10–22 detected Mg + Si + Cu atoms slightly increased with increasing natural ageing up to 24 h. In addition, from the simulated cluster size distribution histogram in Figure 3b and Figure 4b, one can see that there are a few clusters containing more than 15 atoms in Figure 3b (NA sample), while the clusters contain more than 15 atoms increase in Figure 4b (AA sample). Therefore, it can be indicated that the approximate number of atoms in small clusters is less than 15, and that in large clusters it is over 15. Above all, the parameters for the simulation were reasonable as proven by the fact that the simulation was in good agreement with the reported experimental results.

From Figure 3d and Figure 4d, one can see the number of clusters increased continuously until it reached a plateau during NA, while it increased rapidly at beginning and then decreased during AA. Although the number of clusters changed regularly as a whole, there were some partial fluctuations and irregularities, especially in the AA process. The experimental studies of NA and AA in different 6000-series aluminum alloys have similar results [30]. Marceau et al. [30] suggested that about 10–20% of the solute atoms in 6111 aluminum alloy would form clusters during NA, and the number density of Mg-Si clusters increased continuously until reaching equilibrium within two weeks, the size of the clusters was less than 50 (expressed by the number of solute atoms). Esmaeili et al. [31] carried out APT studies of 6111 aluminum alloy between 333 K and 453 K, and Mg-Si clusters were observed. With the increase of aging temperature, the number density of small clusters decreased, the second kind of large clusters appeared, and the average Mg/Si ratio of the clusters increased. These experimental results on clustering were obtained from contingency tables based on APT results. In addition, it was difficult to observe clustering in situ by APT during aging due to the complexity of cluster analysis. The KMC simulation can directly monitor cluster behavior. It can be clearly seen in Figure 5 that the vacancy made small clusters merge into large clusters during AA, as shown in marker A to A1. One can also see some clusters were decomposed at the same time, as shown in marker B to B1, and the small clusters were rich in Si or Mg, while the large clusters had similar Mg and Si contents. From Figure 6, one can see clusters merging and decomposing repeatedly in the simulated AA process, and it was firstly found that the clusters in the Al-Mg-Si-Cu alloy merged and decomposed simultaneously during aging. Therefore, the decrease in the number of clusters in the alloy was attributed to the merging or coalescence and growth of small clusters. The fluctuations and irregularities of the number of clusters were caused by the repeated merging and decomposition of clusters during aging. The rapid increase of the number of clusters was attributed to migration of the initial vacancies (quenching vacancies). Cluster merging and decomposing were both carried out by the captured vacancies. The cluster here was the aggregation of solute atoms, so it was unstable before transforming into higher-order precipitates, such as β″ phase. The baking softening (reversion effect) of 6000-series aluminum alloys also confirmed the cluster decomposition. Baking softening means that the hardness of Al alloys decreases when aged in the temperature range of 443 K to 573 K for a few minutes, and the decrease in hardness was attributed to the decomposition of clusters. The subsequent increase in hardness was attributed to the re-formation of large clusters and the β″ phase transformed from the clusters [32,33,34].

### 3.2. Reversion Ageing Treatment

The reversion ageing (RA) treatment and the aging treatment are two opposite processes. The former is also a possible way to study the early aging behavior [14]. The temperature range for RA in 6000-series aluminum alloys is usually from 473 K to 573 K. In these temperatures, the aged alloy will be softened significantly within several seconds or minutes [34]. Figure 7 shows the hardness change of T4 and T4P samples during RA treatment (treated in an oil bath at 523 K). One can see that the hardness of T4 and T4P samples decreased obviously after RA for tens of seconds, and maximum reductions were 18 HV and 15 HV, respectively. As the RA time is prolonged, one can see that the hardness increases. However, there were some irregularities in the hardness variation. Combining the RA results with the KMC simulation above, the irregularities can be attributed to the complex interaction between the merging and decomposition of clusters due to the fact that RA and AA are opposite processes. The difference between AA and RA was that the decomposition of clusters was dominant during RA, while the merging of clusters was dominant during AA. From Figure 7, one can see that the decrease in hardness of T4 sample was more irregular than that of the T4P sample. The hardness of the T4 sample still decreased after RA for 5 min, while the hardness of the T4P sample began to increase after RA for 2 min. The main reason for the differences was that there were more small clusters/bad clusters in the T4 sample than the T4P sample due to the fact that the artificial aging as part of the T4P is typically done at a temperature within a given range (333 K to 403 K) that can effectively promote the formation of large clusters/good clusters. Precisely, the purpose of T4P temper (pre-aging treatment) has always been to obtain, as much as possible, more large-size clusters than those obtained in the T4 temper [6,28,31,33]. In addition, small clusters were more unstable and easier to be decomposed, while large clusters/good clusters were easier to be transformed into the main strengthening phase, β″ phase. It can be seen from the reports that baking softening usually obviously happened in T4 sample rather than T4P sample, and the researchers suggested that there were more small clusters/bad clusters in the T4 sample and they were decomposed and returned to the matrix during artifact aging, resulting in the decrease in hardness (baking softening) [32,33,34].

Figure 8 shows the hardness of the samples after RA and subsequent NA for 2 weeks and baking. One can see that the hardness of the sample after RA increased during subsequent NA, but it was still lower than that of the sample before RA. The baking hardening (BH) of the samples after RA for 30 s~1 min was improved, and the BH of the samples after RA for 50 s was the best. Figure 8c shows the comparison of BH between the T4 sample and the sample after RA for 50s (R50s). One can see that the hardness of the R50s sample before baking was lower than that of the T4 state sample, and the hardness of the R50s sample after baking was higher. Therefore, the R50s sample was more in line with the requirements of automotive sheets because a lower hardness before baking is more beneficial to stamp forming and a higher hardness after baking is more beneficial to the anti-concave property. The better performance of the RA sample was attributed to the fact that the RA treatment dissolved the small size clusters, which were not conducive to BH. According to the formula of vacancy concentration, as shown in Equation (1), the higher the temperature, the higher the concentration. The temperature of RA treatment was lower than that of the solid solution treatment so that the vacancy concentration in RA samples was lower than that in the quenched samples. Cluster formation is mainly influenced by movable vacancies 14, and low vacancy concentration will inhibit natural aging by inhibiting cluster formation, which is helpful to reduce the negative effects of natural aging. Although the BH of the RA sample was better than that of the T4 state sample, it was not as good as that of direct baking after quenching. Because the vacancy concentration in the RA sample was lower than that in the quenched sample.

Figure 9 shows DSC curves of the samples after RA and subsequent NA for 2 weeks. Activation energy of β″ phase before baking can be calculated by Kissinger method [35]. With a constant heating rate *Φ* (*Φ* = d*T*⁄d*t*), the relationship between the transition rate and the activation energy can be obtained from:(10)$$\ln\left[{\left(\frac{dy}{dT}\right)}_{yi}\Phi_{j}\right]=\ln\left[f\left(y\right)k_{0}\right]-\left(\frac{E}{R}\right)\frac{1}{T_{j}}$$
where *y_i_* is the transformation fraction of all reactions occurring at temperature *T_j_* and heating rate *Φ_j_*.

From Figure 10, one can see that a straight line can be fitted with ln[(d*y*/d*T*)*_yi_Φ_j_*] and 1/*T_j_*. Its slope is −(*E*/*R*), where *E* is the activation energy. In this way, the activation energy can be calculated without any special transformation kinetic model. Activation energies of β″ phase in the samples after RA for 40 s, 50 s, 55 s, 60 s were calculated as 113 kJ/mol, 75 kJ/mol, 131 kJ/mol, 117 kJ/mol, respectively, as shown in Figure 10. One can see that the sample with RA for 50 s has the lowest activation energy. This means that it was easier for β″ phase to precipitate in this sample, which is consistent with the experimental result mentioned above, showing that the sample with RA for 50 s has the best BH. The BH of the alloy was mainly influenced by β″ phase, which was transformed from clusters. At the same time, one can see that the activation energy of β″ phase shows no clear pattern with the extension of RA time. This proves once again that the decomposition and formation of clusters occur simultaneously during RA.

## 4. Conclusions

The main results can be summarized as follows:(1)The formation of many small clusters at the beginning of natural aging and artificial aging is mainly caused by the initial vacancies (quenching vacancies). The clusters containing capturing vacancies can merge with surrounding solute atoms and other clusters, and decompose before reaching a stable size.(2)After repeated merging and decomposition, the clusters reach stability. Small clusters are generally rich in Si or Mg, while large clusters have similar contents of Mg and Si. The decrease in the number of clusters in the alloy is attributed to the merging and growth of small clusters. The fluctuations and irregularities of the number of clusters are caused by the repeated merging and decomposition of clusters during aging.(3)In the process of reversion aging treatment, the baking hardening increases at first and then decreases, and the change of activation energy of β’’ phase is the opposite. There are some partial irregularities in the change of the decrease of in hardness and the activation energy of β″ phase with the extension of reversion aging time due to the interaction of decomposition and aggregation of clusters. The bake hardening of the sample after reversion aging in an oil bath at 523 K for 50 s is the best.

## Figures and Tables

**Figure 1 materials-14-04523-f001:**
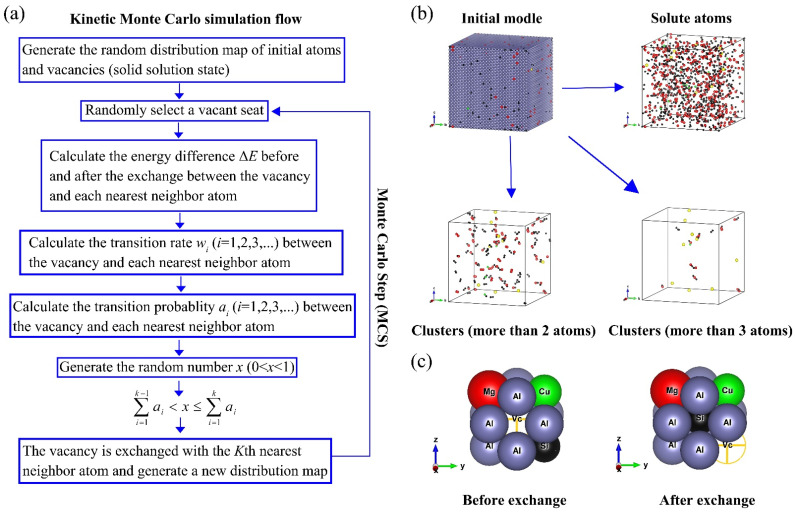
Kinetic Monte Carlo simulation flow and initial model. (**a**) Kinetic Monte Carlo simulation flow; (**b**) Initial model and solute atoms; (**c**) Exchange behavior between vacancy and atoms.

**Figure 2 materials-14-04523-f002:**
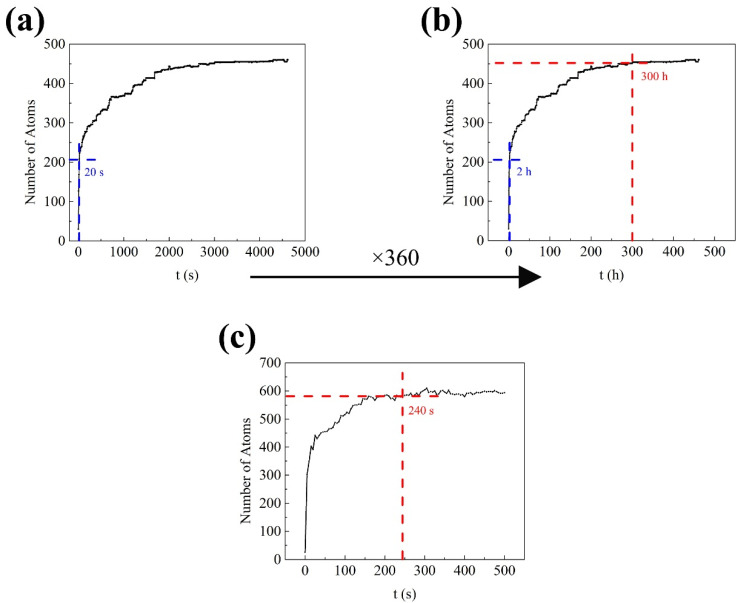
Variation of cluster volume fraction (expressed by the number of solute atoms) during simulated aging of aluminum alloy: (**a**) Natural aging (298 K) before adding the preexponential coefficient of diffusion; (**b**) Nature aging (298 K) after adding the preexponential coefficient of diffusion; (**c**) Artificial aging (448 K).

**Figure 3 materials-14-04523-f003:**
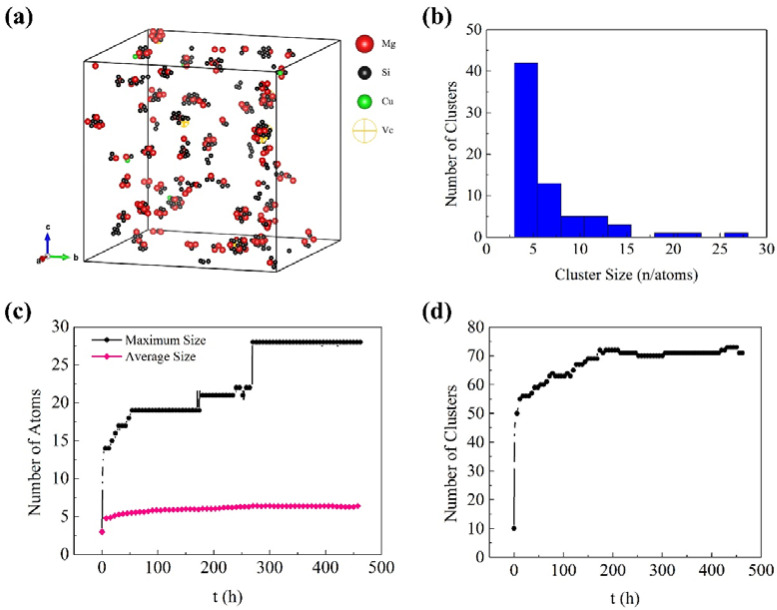
Simulated clusters of aluminum alloy during natural aging (NA): (**a**) 3D morphology of the sample after NA for 300 h; (**b**) Cluster size distribution histogram of the sample after NA for 300 h; (**c**) Variation of cluster size; and (**d**) Variation of the number of clusters.

**Figure 4 materials-14-04523-f004:**
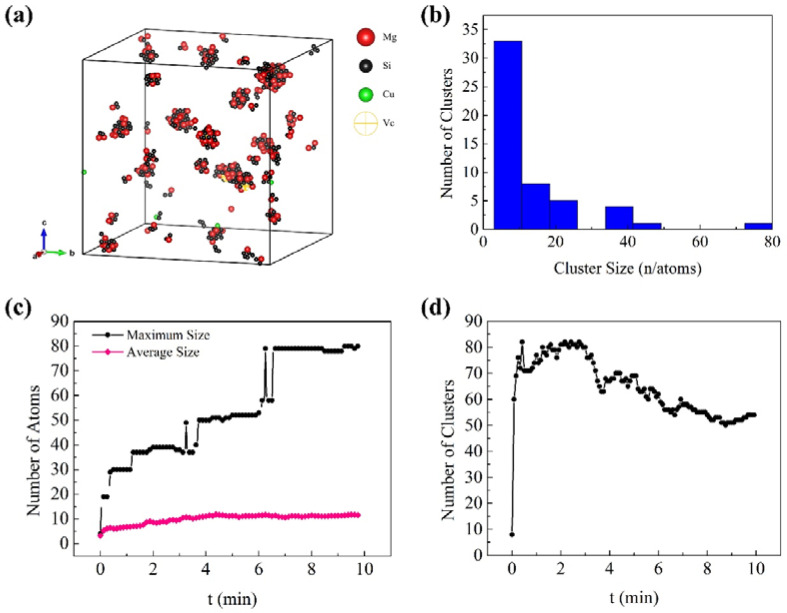
Simulated clusters of aluminum alloy during the artificial aging (AA): (**a**) 3D morphology of the sample after AA for 8 min; (**b**) Cluster size distribution histogram of the sample after AA for 8min; (**c**) Variation of cluster size; and (**d**) Variation of the number of clusters.

**Figure 5 materials-14-04523-f005:**
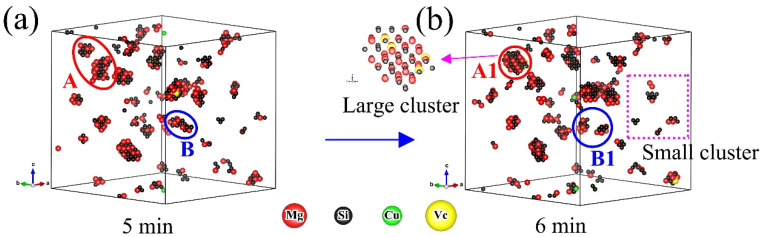
Clusters merging and decomposing simultaneously during artificial aging (448 K): (**a**) 5 min; and (**b**) 6 min.

**Figure 6 materials-14-04523-f006:**
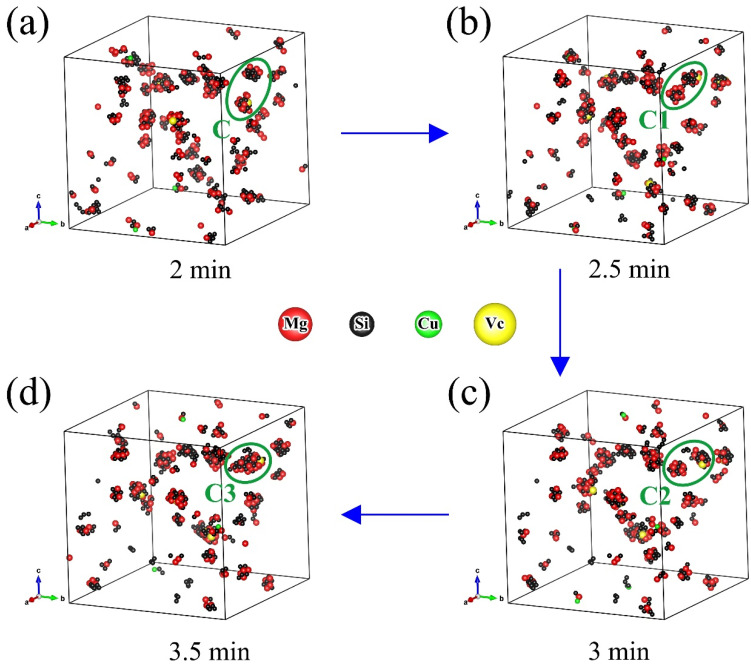
Clusters repeatedly merging and decomposing during artificial aging (448 K): (**a**) 2 min; (**b**) 2.5 min; (**c**) 3 min; and (**d**) 3.5 min.

**Figure 7 materials-14-04523-f007:**
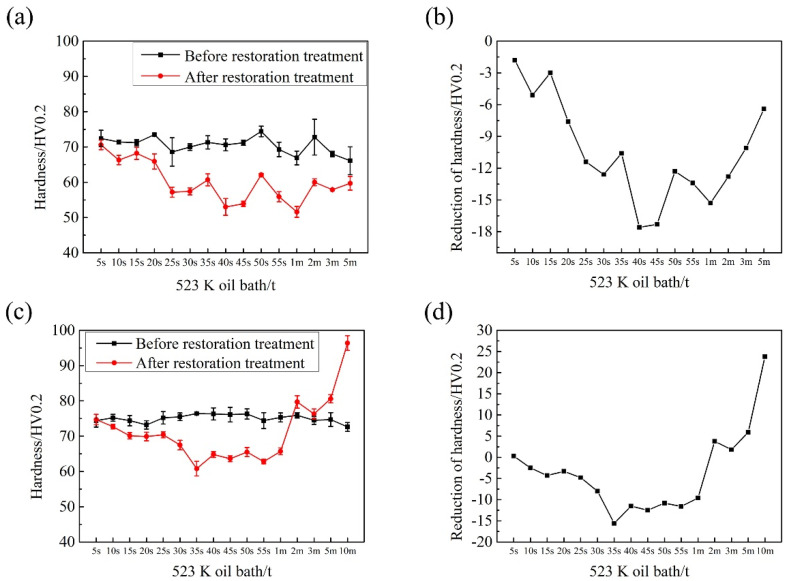
Reversion aging treatment of 6000-series aluminum alloy: (**a**,**b**) T4 temper; and (**c**,**d**) T4P temper.

**Figure 8 materials-14-04523-f008:**
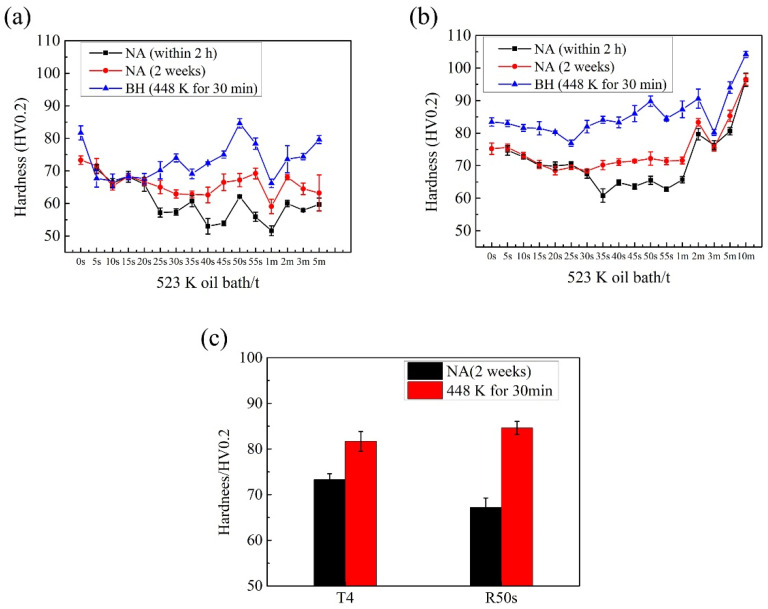
Hardness of the reverse ageing samples after natural aging for 2 weeks and baking hardening: (**a**) T4 temper; (**b**) T4P temper; and (**c**) Baking hardening of T4 and R50s samples.

**Figure 9 materials-14-04523-f009:**
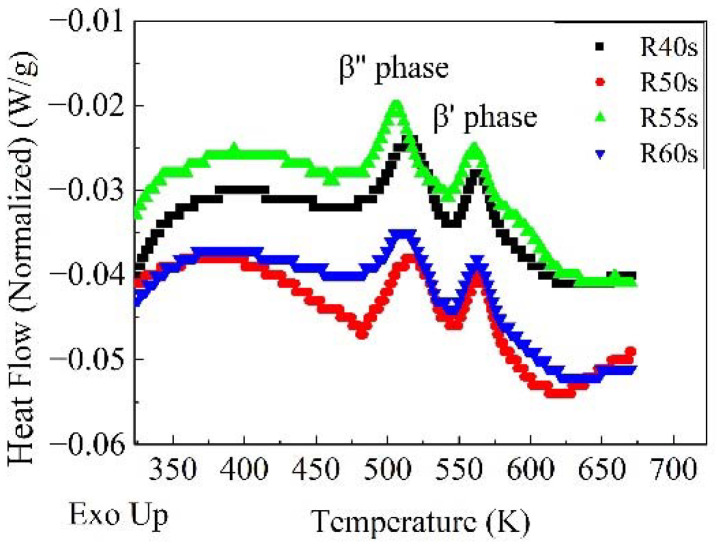
DSC curves of samples after reverse aging and subsequent natural aging for 2 weeks.

**Figure 10 materials-14-04523-f010:**
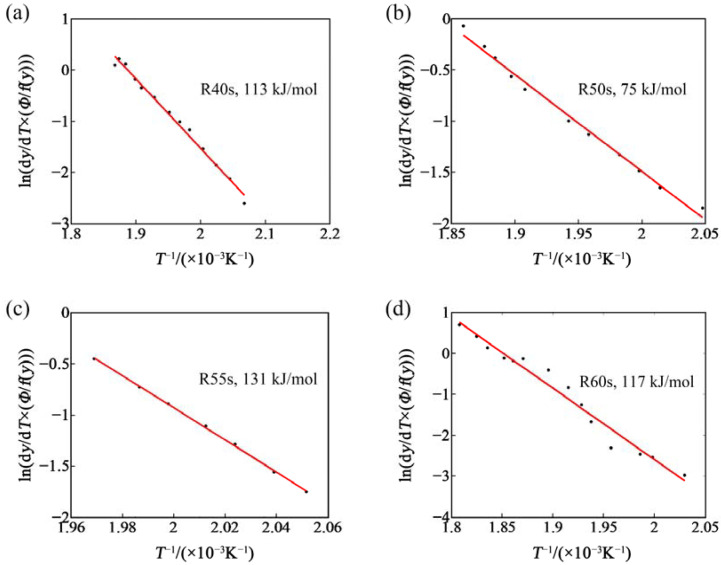
Activation energy of β″ phase before baking: (**a**) R40s, 113 kJ/mol; (**b**) R50s, 75 kJ/mol; (**c**) R55s, 131 kJ/mol; and (**d**) R60s, 117 kJ/mol.

**Table 1 materials-14-04523-t001:** Diffusion parameters of solutes in pure Al [20,21,22,23].

Element	Cohesive Energy (kJ/mol)	Diffusion Constant *D*_0_ (m^2^/s)	Activation Energy *Q* (kJ/mol)	Vacancy Formation Energy Δ*H_v_^f^* (kJ/mol)	Solute-Vacancy Binding Energy *E^b^_i-v_* (eV)
Al	327.279	1.76 × 10^−5^	126.4	67.54	-
Mg	145.643	6.23 × 10^−6^	115.0	55.96	3.86
Si	445.783	2.48 × 10^−4^	137.0	385.94	2.89
Cu	336.236	6.54 × 10^−5^	136.0	119.64	4.34

**Table 2 materials-14-04523-t002:** Interatomic and interatom-vacancy interaction potential parameters (kJ/mol).

	Al	Mg	Cu	Si	Vacancy
**Al**	−54.5	−34.5	−49.3	−50.8	−21.9
**Mg**	-	−16.8	−32.2	−38.1	−10.5
**Cu**	-	-	−48.5	−45.3	−15.8
**Si**	-	-	-	−52.2	−25.9
**Vacancy**	-	-	-	-	0

## Data Availability

The data underlying this article will be shared on reasonable request from the corresponding author.

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
