# Peer review of "Kinetic Monte Carlo Simulation of Clustering in an Al-Mg-Si-Cu Alloy"

_materials, 2021, doi:10.3390/ma14164523_

Round 1
Reviewer 1 Report
The authors have performed some KMC simulations on the cluster formation inside Al-Mg-Si-Cu alloys in order to understand the impact of this cluster on the mechanical properties for automotive applications. Though this work seems to have been done seriously, some points have to be precised to be understandable for a wide range of readers.
The authors seems to have done the simulation on the 6000 series alloys. Is it right ? If yes precise the reference, if not precise the composition of the alloys studied in this work.
In the same way, how the built their initial system to do the KMC ?
If they have studied further composition, the chemical nature of the alloys has an impact on the obtained results.
Another thing which can be perturbating for the reader, with the same initial given set of parameters, they realize simulations ranging from few seconds to further hundreds hours of aging. Is it the real simulation time or the authors know that for an given time of aging there is change in the properties of the alloys which could correspond to a given concentration of defects, for instance ?
How have been obtained determined H and S from previous MD or first-principles calculations, from experimental data ?
All these data have to be clarified to be able to juge the reliability of the calculation and to make the paper more comprehensible.
A little detail, in the first paragraph of the introduction there are further line break in the middle of a sentence.
Author Response
Dear editor and reviewers,
Thank you very much for your kind comments concerning our manuscript entitled "Kinetic Monte Carlo Simulation of clustering in an Al-Mg-Si-(Cu) Alloy". All of these comments are valuable and very helpful for improving this paper as well as for guiding our future research. We have revised the manuscript accordingly. The revised portions are marked in the revised manuscript and a point-to-point response is given in the attachment.

Reviewer 2 Report
please, check the attached pdf

Author Response

(The authors gave the same response as above.)

Round 2
Reviewer 1 Report
The authors have well understood the comments of the reviewer and corrected the manuscript in their direction. In its present form, the quality of the manuscript is highly improved, and, this paper deserves to be published as is.
Author Response
Dear editor and reviewers,
Thank you again very much for your kind comments and recognition of our work.
Yours sincerely,
B. Yang
Q. Ye
Reviewer 2 Report
please, see the attached pdf

Author Response
Dear editor and reviewers,
Thank you again very much for your kind comments concerning our manuscript entitled "Kinetic Monte Carlo Simulation of clustering in an Al-Mg-Si-(Cu) Alloy". All of these comments are valuable and very helpful for improving this paper as well as for guiding our future research. We have revised the manuscript accordingly. The revised portions are marked in the revised manuscript and a point-to-point response is given in the attachment.
Yours sincerely,
B. Yang
Q. Ye
